# Branched Chain Amino Acid Content and Antioxidant Activity of Mung Bean Tempeh Powder for Developing Oral Nutrition Supplements

**DOI:** 10.3390/foods12142789

**Published:** 2023-07-22

**Authors:** Vanessa Violina, Caecilia Eka Putri, Bibiana Widiyati Lay

**Affiliations:** 1Faculty of Biotechnology, Atma Jaya Catholic University of Indonesia, Jakarta 12930, Indonesia; 2Research Center for Indonesian Spices, Atma Jaya Catholic University of Indonesia, Jakarta 12930, Indonesia

**Keywords:** mung bean tempeh, characteristics, branched chain amino acids, oral nutrition supplement

## Abstract

Mung bean (*Vigna radiata*), a non-soybean legume, is known as one of the vegetable protein sources with 27% protein content. Mung bean also has a high content of essential amino acids, including branched chain amino acids (BCAAs). The use of mung bean for tempeh production presumably increases its nutritional value and functional efficacy, most significantly in protein and BCAA content. This tempeh is further applied for developing modern functional foods such as oral nutrition supplements (ONS). ONS can be used as a substitute for food and emergency food due to its complete nutritional content, as well as to treat malnourished patients. This study was aimed to produce mung bean tempeh powder, to formulate a mung bean tempeh one shot ONS high in BCAA content, and to determine its proximate analysis, antioxidant activity, and sensory characterization. Mung bean tempeh powder was successfully obtained with a yield of 37.50%, protein 39.19%, total amino acids 286.21 mg/g, essential amino acids 117.97 mg/g, and BCAAs 54.14 mg/g. There were 6 ONS formulas that were made with the combination of mung bean tempeh powder, palm sugar or honey, olive oil, and addition of an emulsifier. The selected formulas (F1 and F4) as well as commercial mung bean juice were sensory analyzed by applying an appropriate hedonic test. The results showed that the panelists both liked ONS F1 and F4 (*p* > 0.05). In addition, both ONS F1 and F4 at 5% demonstrated a significant antioxidant capacity, 92.79% and 82.57% of ascorbic acid, respectively. These data suggest that mung bean tempeh containing high branched amino acids could be recommended as a functional ingredient that gives health promotion for ONS development.

## 1. Introduction

Tempeh is one of Indonesia’s most famous traditional foods made from fermented beans. Beans are a potential source of vegetable protein with very low-fat content and contain various vitamins and minerals. The fermentation process can increase nutritional value due to the hydrolysis process of complex compounds in beans into simpler compounds by microorganisms [1]. In addition, it also helps in the removal of anti-nutritional compounds, increases certain nutrients, and the formation of bioactive compounds [2]. Several processes during tempeh fermentation also affect the anti-nutritional compounds in the beans. The soaking, dehulling, cooking, and fermentation processes by *Rhizopus oligosporus* caused the activation of trypsin inhibitory activity and reduction of phytic acid content and total phenolic compounds in soybean, kidney bean, and jojoba seeds [3]. It increases the levels of protein, polyunsaturated fatty acids (PUFA), and antioxidant compounds [4]. In general, tempeh production utilizes soybeans as a main ingredient due to their high protein and BCAA content. However, the rapid increase in soybean consumption has not been matched by an adequate rise in domestic soybean production. The gap has been filled with imported soybeans which fluctuated soybean prices [5]. This has prompted research using other non-soybean legumes as substitute tempeh substrates, namely, common beans, peanuts, lupine beans, adzuki beans, mung beans, pigeon peas, and many more. Mung bean (*Vigna radiata*) is a prime candidate due to its abundant amount in Indonesia and contains 27% vegetable protein with a high content of essential amino acids, including BCAA, whose amount is not inferior to soybeans [6]. Mung beans are the third most widely grown legume in Indonesia after soybeans and peanuts. Regions in Indonesia which produce mung beans are Nanggroe Aceh Darussalam, West Sumatra, South Sumatra, West Java, Central Java, East Java, North Sulawesi, South Sulawesi, West Nusa Tenggara, and East Nusa Tenggara. Mung bean porridge is traditionally known as a popular breakfast food in most regions in Indonesia. Due to its high contents of fiber, carbohydrate, and protein, this porridge is believed to be an importance source of energy for the body and the right choice for breakfast foods. A previous study demonstrated the use of distilled water extraction in a ratio of 1:1 for obtaining protein isolates from mung bean tempeh [7].

Oral nutritional supplements (ONS) are ready-to-drink products existing in liquid, semi-liquid, or solid form that are rich in macro- and micronutrients at various concentration levels. The usage of ONS significantly improves nutritional status and appetite [8]. ONS can be used as a substitute for food and emergency food due to its complete nutritional content, as well as to treat malnourished patients. A shot is a type of ONS made from protein and fat in a concentrated form. The main recipe for making ONS one shot is mung bean tempeh which is high in protein, including branched chain amino acids (BCAAs). BCAAs are a trio of amino acids consisting of leucine, isoleucine, and valine that can only be obtained from food. These BCAAs have been recognized as one of the most promising functional ingredients for health promotion, which includes preventing muscle loss at any age, reducing the negative nitrogen balance in biological systems, and treating patients with cirrhosis and liver cancer [9,10].

Several studies have been conducted to utilize mung bean tempeh as a functional food by Maryam [11] and enteral food formulations based on high-protein soybean tempeh powder by Faidah et al. [12]. However, currently there is no research that utilizes tempeh mung beans which are high in protein and BCAAs to produce functional drinks in the form of ONS one shot. The abundant availability and nutritious content of local mung beans leads this study to produce mung bean tempeh powder, to formulate a mung bean tempeh one shot ONS high in BCAA content, and to determine its proximate analysis, antioxidant activity, and sensory characterization.

## 2. Materials and Methods

### 2.1. Mung Bean Powder and Mung Bean Tempeh Powder Production

Mung bean samples were collected from a traditional market in Kupang (East Nusa Tenggara, Indonesia). Mung bean powder (unfermented) and mung bean tempeh powder (fermented) production were carried out using the methods of Widjajasaputra and Dinar [13,14]; 500 g of mung beans was first prepared by washing them until clean and then soaked overnight at room temperature (30 °C). For the unfermented mung bean powder, after the soaking process, the mung beans were hydrated and dried in an incubator (Memmert INB200-400, Büchenbach, Germany) at 45 °C for 48 h. Dried mung beans were crushed using a food processor (Phillips HR7310 food processor, Amsterdam, The Netherland) and filtered using a 100-mesh sieve (Retsch, Nordrhein-Westfalen, Germany) to obtain mung bean powder. For tempeh mung bean powder preparation, after soaking overnight, mung beans were boiled for 30 min until they were half-cooked. Then, the mung beans’ skin was peeled off and the skinless cooked mung beans were obtained at 823 g. Next, these mung beans were steamed for 10 min. After steaming, the mung beans were then cooled down to room temperature. Subsequently, mung beans were mixed with 1 g of Raprima^®^ tempeh starter (containing *Rhizopus oligosporus* strain; PT Aneka Fermentasi Industri, Bandung, Indonesia) and stirred until they were fully blended. Lastly, the mung beans and yeast mixture were transferred into plastic bags that had been perforated and left between 36 to 40 h at room temperature (30 °C). Mung bean tempeh powder preparation was conducted by following the method derived by Jauhari et al. [15]. Frozen samples of mung bean tempeh were put into a freeze-dryer (Christ Alpha 1-2 LDplus −55 °C freeze dryer, Osterode am Harz, Germany; 0.01 mbar; −60 °C) for 48 h. The moisture content of mung bean tempeh powder after freeze-drying was 7.73%. Afterward, the freeze-dried samples were crushed with a food processor until they achieved a smooth and powdery texture. Furthermore, the powder was then filtered using a 100-mesh sieve.

### 2.2. Proximate Analysis

Proximate analysis for ash, moisture, water, protein, and fat contents was carried out according to the method of the Association of Official Analytical Chemists (AOAC) [16]. Carbohydrate content was calculated by using the difference method, through calculating the percentage difference of total contents of the water, ash, protein, and fat with 100%. 

### 2.3. Amino Acid Analysis

Amino acid analysis for 18 essential and non-essential amino acids was performed using the UPLC system with C18 (1.6 μm particle size, 2.1 × 150 mm column) (ACQUITY UPLC H-Class, Waters, Milford, MA, USA), according to the procedure established by Waters [17]. Amino acid standard concentration points and internal standards were prepared. A 50 mg sample was weighed and put into a 100 mL volumetric flask. Next, HCl solution was then added and homogenized. The solution was filtered with a 0.20 μm syringe filter (Corning^®^, New York, NY, USA) and the filtrate was collected. Standard internal was added, followed by derivatization. Finally, the solution was injected into the UPLC system.

Methionine and cystine were then measured using the LC-MS/MS system with Imtakt Intrada Amino Acid (50 × 3.0 mm column (Shimadzu, Kyoto, Japan) by following the method of Dahl-Lassen et al. [18]. Seven points of internal standards were made in 2 mL vials. The solvent used was a standard solvent; 0.5 g of sample were weighed and transferred into a 20 mL headspace vial, then stored in a cooling water bath at −10 °C for 15 min. The thawing process was carried out until all the sample melted, then sodium bisulfite was added and the vial shook carefully. After three hours, hydrolysis solvent was added. The vial headspace was then uncapped into an open vial and heated in an incubator at 110 °C for 1 h. The vial headspace was then capped and reheated at 110 °C for an additional 21 h. The bottle was then removed, cooled, placed in a 100 mL beaker glass, and diluted to 25 mL with aquadest. The pH was adjusted to 2.20 and then transferred to a 50 mL volumetric flask containing aquadest. Afterwards, the solution was transferred to a 2 mL tube, centrifuged, filtered through an RC 0.20 µm membrane filter (Cytiva, Marlborough, MA, USA), and then injected into the LC-MS/MS system. 

### 2.4. Fatty Acid Analysis

Fatty acid analysis was carried out by using the gas chromatography system with a flame ionization detector (GC-FID; Perkin Elmer, Waltham, MA, USA) following the modified method of Nur et al. [19]. The working standard solution was prepared at one concentration point in hexane solvent; 50 mg of fat sample was placed in a 20 mL screw vial. Methyl tert-butyl ether and transesterification solution were added, then vortexed for 10 s. After the homogenization process, the sample was centrifuged with hexane and neutralizing solution. The organic phase was collected in 2 mL vials and injected into a GC-FID system. 

### 2.5. ONS Formulation

ONS formulation was carried out by following the ESPEN guidelines as described by Weimann et al. [20]. Mung bean tempeh powder was used as a source of protein, palm sugar and honey were taken as a source of carbohydrates, and olive oil was used as a source of fat in this formulation. This formulation was made by mixing mung bean tempeh powder with palm sugar or honey and water in a beaker glass, then combining them with heated fat and an emulsifier. This formulation was made in the form of a shot of 50 mL per serving. Table 1 showed six formulations of mung bean tempeh ONS formulation with various compositions. These formulations were tested by the preliminary sensory test with 5 parameters (color, aroma, taste, viscosity, and overall) for 3 days. The acceptable formulas were selected for further testing by sensory test.

### 2.6. Antioxidant Activity Assay

Antioxidant activity was analyzed using 1,1-diphenyl-2-picrylhydrazyl (DPPH), following the method of Ghimeray et al. with a slight modification [21]. The tested samples were mung bean tempeh powder and ONS formulations of mung bean tempeh (F1 and F4). Mung bean tempeh powder was extracted using buffer phosphate 50 mM pH 7.0 and diluted to various concentrations (0.5%, 1%, and 5% *w*/*v*) in methanol. For ONS formulations, they were also diluted to these concentrations in methanol; 100 μL of each sample with different concentrations was added with 100 μL of 0.6 mM DPPH solution (in methanol). Methanol was used as a blank of the samples; meanwhile, ascorbic acid was used as a positive control. Each solution was vortexed and incubated in the dark for 30 min. Solutions were pipetted into 96 wells of microplates and absorbance was measured at 517 nm using a microplate reader (INNO-M, LTEK Gyeonggi-do, Republic of Korea). This analysis was performed in triplicate. Antioxidant activity was then calculated using the following formula: Antioxidant = [*A*blank − *Asample*/Ablank] × 100%. Ablank denotes the absorbance of the control reaction, while Asample denotes the absorbance in the presence of the sample.

### 2.7. Sensory Test

The sensory test was carried out by hedonic test following the method of Meilgaard et al. [22]. This hedonic test involved 35 untrained panelists (20 women and 15 men) with an age range of 19–23 and used three samples of ONS shot, including F1, F4, and CF (commercial mung bean juice; ABC Minuman Sari Kacang Hijau by PT Heinz ABC Indonesia, Karawang, West Java, Indonesia). This study was ethically approved by the committee of the Research Centre for Indonesian Spices, Atma Jaya Catholic University of Indonesia (003/PRRN/04/2022). Written informed consent for this study was obtained from panelists. The panelists were each given two samples of mung bean tempeh nutritional supplements with different formulas and one commercial mung bean juice. Each sample was named with three random digit codes. Panelists were then given time to evaluate each sample based on five parameters including color, aroma, taste, viscosity, and overall on a scale of 1–7 which determined 1 as very disliked and 7 as very liked. 

### 2.8. Statistical Analysis

Quantitative measurements were reported as mean ± standard error mean (SEM) from proximate, amino acid, fatty acid, antioxidant, and sensory tests. All experiments were conducted in triplicate. Analyses were performed with SPSS software using one-way analysis of variance (ANOVA), followed by the Tukey test to determine the significant variation in results with a 5% significance level.

## 3. Results

### 3.1. Mung Bean Tempeh Powder

The results of making tempeh from mung bean substrate can be observed in Figure 1a. Mung bean tempeh powder was successfully obtained through freeze-drying, powdering with a food processor, and filtering using a 100-mesh sieve (Figure 1b). This process produced 187.50 g mung bean tempeh powder from 500 g of dried mung beans, with a yield of 37.50%.

### 3.2. Proximate Content of Mung Bean Tempeh Powder

Table 2 indicates the proximate results of mung bean tempeh powder and mung bean powder. The protein content is 39.20% in mung bean tempeh powder, which is 1.70 times higher than the non-fermented mung bean powder. The total fat from tempeh mung bean powder increased by 3.60-fold to 33.00%. On the other hand, fermentation decreased the amount of carbohydrates in mung bean tempeh powder by 1.40-fold (to 47.10%). Ash and moisture content in mung bean tempeh powder were measured at 2.80% and 7.70%, respectively.

### 3.3. Amino Acid Profiles of Mung Bean Tempeh Powder

Table 3 shows that the increase of amino acids, essential amino acids, and BCAAs was approximately twice in mung bean tempeh powder compared to raw mung bean powder. The concentration of BCAAs in mung bean tempeh powder and mung bean powder represented the 45.89% and 41.89% of the content in essential amino acids, respectively. The highest amino acids found in mung bean tempeh powder were glutamic acid and alanine, while in mung bean powder were glutamic acid and aspartic acid.

### 3.4. Fatty Acid Profiles of Mung Bean Tempeh 

The fatty acid content in mung bean tempeh is illustrated in Table 4. The mung bean tempeh powder was characterized by a high content of unsaturated fatty acids (UFA, 2.92%) with the essential omega-6 being the most abundant chemical class of fatty acids. 

### 3.5. Sensory Evaluation Results

A preliminary sensory test for six ONS formulas containing mung bean tempeh powder was conducted over three days at room temperature. This test was performed in five sensory parameters, including color, viscosity, taste, scent, and overall to find the acceptable formula (Table 5 and Figure 2). The overall parameters determined the acceptable formulas based on their stability. The accepted formulas (F1 and F4) were continued for sensory evaluation by the panelists. Table 6 and Figure 3 show the results of F1, F4, and control (CF) sensory analysis on the five parameters tested, including color, taste, aroma, viscosity, and overall. In the parameters of aroma and viscosity, there was no significant difference between these data obtained from the samples tested (*p* > 0.05). On the other hand, color, taste, and overall parameters showed a significance difference (*p* < 0.05). The CF sample was preferred by panelists based on the taste and overall parameters. Meanwhile, the F4 sample was preferred based on the color parameter.

### 3.6. ONS Calories and Nutrition

Table 7 shows the final nutritional content of mung bean tempeh ONS one shot formulas (F1 and F4). In one ONS serving (50 mL) of F1, the amount of protein, carbohydrates, fat, and calories were 2.04 g, 7.65 g, 2.07 g, and 56 kcal, respectively. Whereas ONS F4 used a different type of sweetener, then, the amount of carbohydrates differs to 6.42 g. The calorie content of macromolecules (protein, carbohydrates, and fat) in both formulations of ONS complies with ESPEN guidelines (Table 8).

### 3.7. Antioxidant Activity of Mung Bean Tempeh Powder and ONS

Mung bean tempeh powder and ONS formulas showed a consistent higher antioxidant activity with increasing concentrations (Figure 4). Our data revealed that both ONS F1 and F4 tested at 1% evidenced a not significant difference in terms of antioxidant activity compared to the ascorbic acid standard. Moreover, at 5%, ONS F1 and F4 exerted a considerable antioxidant capacity, 92.79% and 82.57% of ascorbic acid, respectively. Meanwhile, mung bean tempeh powder at 5% showed the lowest antioxidant activity (61.84%).

## 4. Discussion

This study intended to produce mung bean tempeh powder, beginning with the process of making mung bean tempeh. Kupang’s mung beans were chosen as a substrate to utilize locally sourced Indonesian mung beans. The fermentation of mung beans into tempeh improves its nutritional contents including essential amino acids, forms bioactive compounds that are beneficial to health, and enhances organoleptic properties [23]. The mung bean tempeh fermentation process was carried out for 36 to 40 h. Over-fermentation will result in protein degradation, which increases ammonia levels and gives the tempeh a strong malodorous odor [24]. The production of mung bean tempeh powder was achieved by using a freeze-dryer and food processor. Freeze-drying is a more beneficial food drying technique that could maintain both the aroma and color of food products [25]. The yield obtained in the production of mung bean tempeh powder was 37.50% (39.19% of protein), almost equivalent to ±40.00% yield of soybean tempeh powder (45.55% of protein) reported by Jauhari et al. [15]. Another alternative method to make tempeh powder (soybean) can be completed through a natural process, which is drying it in the sun which gives a yield of 31.52% (16.33% of protein) [26].

For nutritional values, proximate profiles showed that protein content in mung bean tempeh powder was up to 1.7 times higher compared to that of mung bean powder only (Table 3). These data also aligned with the previous report composed by Lestari et al. [27]. Another report showed that the amount of protein in soy tempeh powder was 1.20 times higher than mung bean tempeh powder [14]. Fermented mung beans implied an increase of 2× in amino acid content as compared to raw mung beans (Table 4). The BCAA content in mung bean tempeh was 45.89% of the total essential amino acids, comparable to a slightly lower content at 41.89% in unfermented mung beans. Mung bean tempeh contains 54.14 mg/g BCAAs which is significantly higher than the BCAA content found in unfermented mung beans (38.80 mg/g) reported by Lee et al. [28]. Overall, fermentation is proven to enhance the nutritional value of raw mung beans in terms of the total amino acids, essential amino acids, and BCAAs. BCAAs have been used as one of the dietary supplements in various pathophysiological conditions. Supplementing with BCAAs can prevent muscle loss at any age, reduce the negative nitrogen balance in biological systems, and treat patients with cirrhosis and liver cancer [10].

In mung bean tempeh powder, the highest amino acid content was glutamic acid, which was 48.76 mg/g, followed by alanine, which was 37.39 mg/g (Table 3). Glutamic acid is a non-essential amino acid found in abundant quantities within mung beans. Glutamic acid in the body turns into glutamate which plays a role in the formation of proteins in the body, assists brain nerve cells to send and receive information from other cells, and plays an important role in memory, cognition, and mood regulation [29]. Meanwhile, alanine acts as a source of energy for muscle tissue, boosts the body’s immune system by producing antibodies, and assists in sugar metabolism within the body [30]. In addition, lipid content in mung bean tempeh powder was 3.28%, below 5% (Table 3). Meanwhile, in the fatty acid profiles shown in Table 5, the levels of omega-3, omega-6, and omega-9 are below 2%. Therefore, it was concluded that mung bean tempeh powder was high in protein but did not have a high fat content.

Preliminary testing was performed to discover the two best formulations for sensory testing through a hedonic test. As a result, F1 and F4 one shot ONS formulas were chosen to be the best formula based on their stability and sensory parameters within 3 days of observation (Table 6 and Figure 2). Meanwhile, with the other non-selected formulations, separation of the water and oil phases occurred due to the unmixed and homogenous composition. F1 and F4 were made from mung bean tempeh powder, palm sugar (F1) or honey (F4), olive oil, and addition of an emulsifier. Sensory testing with seven-point hedonic scales showed that there was a significant difference in the color parameter between samples F1 and CF towards F4 with a *p*-value < 0.05, meaning the panelists preferred samples F4 in color rather than F1 and CF. In taste and overall parameters, a significant difference (*p* < 0.05) was shown by samples F1 and F4 towards the CF sample, meaning the CF sample was the most preferred compared to the other samples. To conclude, F1 and F4 samples were not significantly different within their sensory parameters, excluding the color (Table 7 and Figure 3). Therefore, the F1 and F4 samples were chosen to be the final recipe for the mung bean tempeh ONS shot. Sensory evaluation was utilized as a guide to achieve the desired product, identifying areas for improvement, determining whether optimization has been achieved, and assessing consumer acceptance and preference [31].

Nutritional content in the mung bean one shot followed the ESPEN guidelines. One serving (50 mL) of mung bean tempeh ONS consisted of 2.04 g protein, 7.65 g (F1) or 6.42 g (F4) carbohydrates, and 2.07 g fat. F1 contained 14.30% kcal of protein, 53.50% kcal of carbohydrates, and 32.10% kcal of fat (Table 7). While energy of protein, carbohydrates, and fat found in F4 were 15.60% kcal, 49.20% kcal, and 35.20% kcal, respectively (Table 8). The energy which came from protein, carbohydrates, and fat in F1 and F4 complied with ESPEN standards which suggest 8–30% of energy comes from protein, 30–65% kcal from carbohydrates, and 25–40% kcal comes from fat [32].

The presence of antioxidants in fermented food products, especially mung beans, has been widely reported [33]. Fermented food products derived from plant-based foods have high-quality protein with plenty and complete amino acid contents. Several amino acids in the form of tryptophan, methionine, histidine, lysine, cysteine, arginine, and tyrosine are reported to have a high antioxidant capacity [34]. In this study, antioxidant tests were carried out on samples of mung bean tempeh powder and two ONS (F1 and F4). Interestingly, both ONS F1 and F4 tested at 1% evidenced a not significant difference in terms of antioxidant activity compared to the ascorbic acid standard. Meanwhile, ONS F1 and F4 at 5% possessed a considerable antioxidant capacity, 92.79% and 82.57% of ascorbic acid. Mung bean tempeh powder also showed potential antioxidant activity (61.86%). This indicates that the mung bean tempeh has a good antioxidant effect. The antioxidant activity of mung bean tempeh was the same as soybean tempeh (52.72–67.61%), discovered by Barus et al. [35]. The higher antioxidant activity in ONS F1 and F4 is caused by the addition of palm sugar, honey, and olive oil which could increase the antioxidant effect. 

The high-protein ONS shot contains BCAAs (isoleucine, leucine, and valine) as well as nutrients such as glucose and lipids. They both offer a rich diet of essential nutrients that can be used as an alternative adjunct to treat acute malnutrition. Additionally, BCAAs also claim to enhance protein synthesis and attenuate inflammation [36]. The recommended BCAA intake is 40, 20, and 19 mg/kg/day of leucine, valine, and isoleucine, respectively [34]. BCAAs in the ONS shot was 281.53 mg per 50 mL. This ONS can be used as an additional supplement to meet BCAA intake with the amount of consumption adjusted to the needs according to body weight. Multiple nutrient deficiencies caused by severe acute malnutrition leads to severe oxidative stress [37]. This effect can be mitigated by administering antioxidant supplements, including this ONS drink which has been proven to have high antioxidant activity. Further research is required before using the high-antioxidant ONS mung bean tempeh drink as an adjunctive therapy to treat this disease and minimize the risk.

Meal replacements are popular alternative food products. Meal replacements are used to replace regular meals by providing complete nutrition. According to The European Commission, meal replacements must contain 200–250 kcal of energy, no more than 35% fat content, and 25–50% protein content [38]. To qualify as a meal replacement, four servings of ONS mung bean tempeh must be consumed with a calorie content of 224 kcal, energy from fat by 32%, and energy from protein by 14% for F1. Whereas F4 contains 208 kcal, 35% fat content, and 15.6% protein content. From these data, mung bean tempeh is still lacking in protein content, therefore is unable to act as a meal replacement. A possible method to achieve the desired protein content is by increasing the amount of mung bean tempeh powder, which is the main source of protein in this formula.

## 5. Conclusions

Mung bean tempeh powder was successfully obtained with a yield of 37.5%, protein 39.19%, a total of 286.21 mg/g amino acids, essential amino acids 117.97 mg/g, and 54.14 mg/g of BCAAs. Tempeh powder at 5% also had good antioxidant activity (61.84%). There were two formulas of ONS one shot mung bean tempeh (F1 and F4). Both ONS F1 and F4 were equally liked by the panelists with a BCAA content of 281.53 mg per 50 mL. Both ONS F1 and F4 at 5% showed significant antioxidant activity, 92.79% and 82.57% of ascorbic acid, respectively. These data suggest that mung bean tempeh containing high branched amino acids could be recommended as a functional ingredient that gives health promotion for ONS development.

## Figures and Tables

**Figure 1 foods-12-02789-f001:**
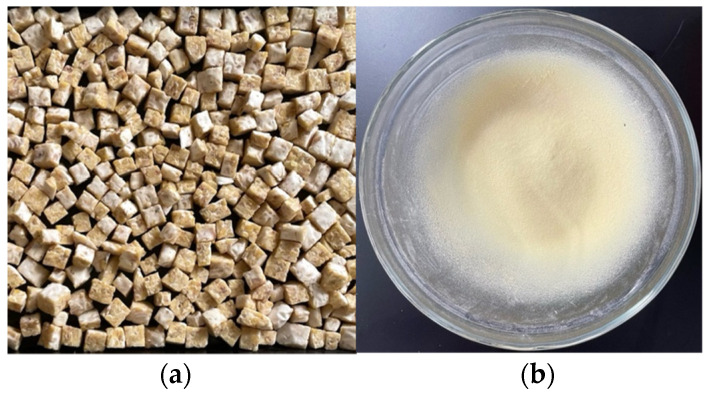
Fresh mung bean tempeh (**a**) and mung bean tempeh powder (**b**).

**Figure 2 foods-12-02789-f002:**
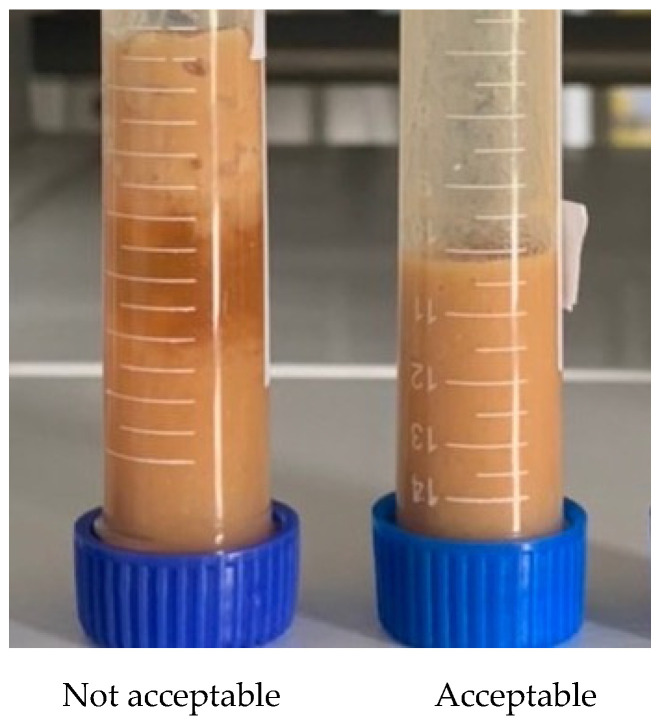
Mung bean tempeh ONS shot formulas.

**Figure 3 foods-12-02789-f003:**
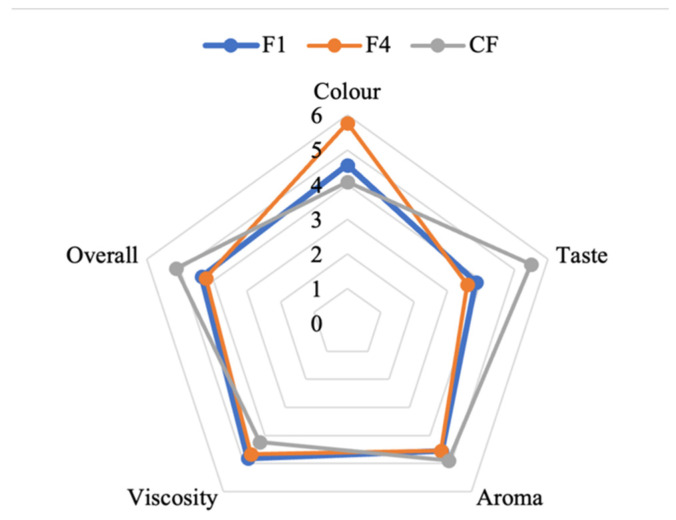
Sensory evaluation results of mung bean tempeh ONS shots. A seven-point hedonic scale is used to rate five sensory attributes.

**Figure 4 foods-12-02789-f004:**
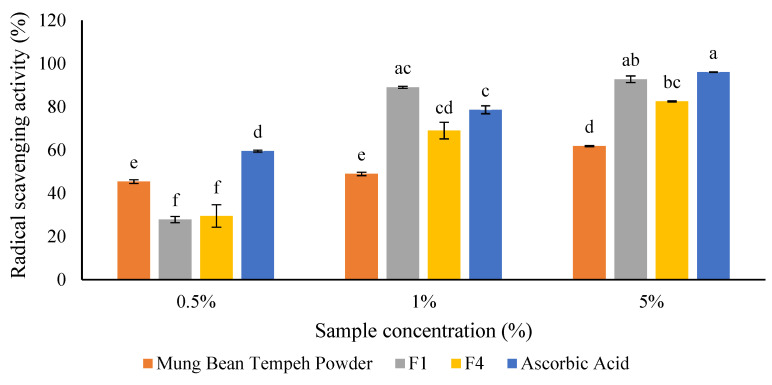
Radical scavenging activity of mung bean tempeh powder and its ONS products at various concentrations. Ascorbic acid was used as the reference. The letter of (a) was significantly different with (b), (c), (d), (e), and (f). Similar letters were not significantly different (*p* > 0.05). All experiments were conducted in triplicate.

**Table 1 foods-12-02789-t001:** Mung bean tempeh ONS formulations.

Ingredients	Content (%)					
	F1	F2	F3	F4	F5	F6
Mung bean tempeh powder	10.40	7.60	6.00	10.40	7.60	6.00
Palm sugar	10.40	7.60	6.00	-	-	-
Honey	-	-	-	10.40	7.60	6.00
Olive oil	3.70	2.70	2.10	3.70	2.70	2.10
Emulsifier SP	1.50	1.10	0.90	1.50	1.10	0.90
Water	74.00	81.00	85.00	74.00	81.00	85.00

**Table 2 foods-12-02789-t002:** Proximate content in mung bean tempeh powder.

Parameter	Unit	Mung Bean Tempeh Powder	Mung Bean Powder
Ash content	%	2.76 ± 0.06	3.21 ± 0.01
Moisture content	%	7.73 ± 0.18	7.29 ± 0.01
Protein content	%	39.19 ± 0.50	23.60 ± 0.47
Carbohydrate (by difference)	%	47.06 ± 0.28	65.00 ± 0.48
Total fat	%	3.28 ± 0.11	0.91 ± 0.01
Calorie from fat	Kcal/100 g	29.48 ± 0.95	8.15 ± 0.06
Total calories	Kcal/100 g	374.46 ± 0.08	362.55 ± 0.01

**Table 3 foods-12-02789-t003:** Amino acids content in mung bean tempeh powder.

Levels	Amino Acids	Mung Bean Tempeh Powder (mg/g)	Mung Bean Powder (mg/g)
Overview	Total amino acids	286.21 ± 0.58	152.25 ± 0.36
	Total essential amino acids	117.97 ± 0.22	65.94 ± 0.13
	Total BCAAs	54.14 ± 0.05	27.62 ± 0.05
Higher levels	Glutamic acid	48.76 ± 0.11 ^a^	29.22 ± 0.10 ^a^
	Alanine	37.39 ± 0.14 ^b^	7.61 ± 0.02 ^b^
	Aspartic acid	23.37 ± 0.04 ^c^	18.44 ± 0.05 ^c^
	Leucine *	22.97 ± 0.03 ^d^	13.04 ± 0.04 ^d^
	Phenylalanine *	19.99 ± 0.05 ^e^	10.60 ± 0.03 ^e^
	Lysine *	17.81 ± 0.06 ^f^	12.03 ± 0.02 ^f^
	Valine *	17.21 ± 0.03 ^g^	7.99 ± 0.01 ^g^
	Isoleucine *	13.96 ± 0.01 ^h^	6.58 ± 0.01 ^h^
	Arginine	13.50 ± 0.01 ^i^	1.07 ± 0.01 ^i^
	Threonine *	13.50 ± 0.02 ^i^	5.93 ± 0.01 ^j^
	Glycine	12.48 ± 0.05 ^j^	6.71 ± 0.01 ^h^
	Histidine *	12.37 ± 0.00 ^j^	4.68 ± 0.02 ^k^
	Tyrosine	11.92 ± 0.04 ^k^	3.93 ± 0.00 ^l^
	Serine	11.67 ± 0.01 ^l^	9.24 ± 0.03 ^m^
	Proline	9.14 ± 0.01 ^m^	7.13 ± 0.01 ^n^
Lower levels	Tryptophan *	0.17 ± 0.03 ^n^	1.68 ± 0.01 ^o^
	Methionine	0.01 ± 0.00 ^n^	3.39 ± 0.00 ^p^
	Cystine	ND	2.90 ± 0.00 ^q^

* Indicates essential amino acid; ND, not detected. Data were expressed as mean value. Different letters for each analysis indicate significant differences (*p* < 0.05). All experiments were conducted in triplicate.

**Table 4 foods-12-02789-t004:** Fatty acids content in mung bean tempeh powder.

Parameters	Chemical Structure	Fatty Acids (%)
Individual fatty acids		
Linolenic acid	C18:3	0.39 ± 0.01 ^a^
Linolenic acid	C18:3 ω3	0.23 ± 0.00 ^b^
Linolenic acid	C18:3 ω6	0.16 ± 0.00 ^c^
Linoleic acid	C18:2	1.48 ± 0.00 ^d^
Linoleic acid	C18:2 ω6	1.48 ± 0.00 ^d^
Oleic acid	C18:1 ω9 C	1.01 ± 0.00 ^e^
Stearic acid	C18:0	0.12 ± 0.00 ^f^
Heptadecanoic acid	C17:0	0.02 ± 0.00 ^g^
Palmitoleic acid	C16:1 Ω7	0.02 ± 0.00 ^g^
Palmitic acid	C16:0	0.73 ± 0.00 ^h^
Myristic acid	C14:0	0.02 ± 0.00 ^g^
Sums of fatty acids		
SFA		0.87 ± 0.00
MUFA		1.05 ± 0.00
PUFA		1.87 ± 0.00
UFA		2.92 ± 0.00
Omega-3		0.23 ± 0.00
Omega-6		1.64 ± 0.00
Omega-9 Omega-7		1.01 ± 0.00 0.02 ± 0.00

Data were shown in mean ± standard deviation. Different letters for each analysis indicate significant differences (*p* < 0.05). All experiments were conducted in triplicate. SFA, saturated fatty acid; MUFA, medium unsaturated fatty acid, PUFA, polyunsaturated fatty acid; UFA, unsaturated fatty acid.

**Table 5 foods-12-02789-t005:** Preliminary testing of mung bean tempeh ONS shot formulas.

Formulation	Day	Parameter				
		Color	Viscosity	Taste	Scent	Overall
F1	H0	Dark brown	Slightly thick, smooth	Moderate sweet, tempeh	Strong tempeh and palm sugar	Acceptable
	H1	Dark brown	Slightly thick, smooth	Moderate sweet, tempeh	Strong tempeh and palm sugar	
	H2	Dark brown	Slightly thick, smooth	Moderate sweet, tempeh	Strong tempeh and palm sugar	
	H3	Dark brown	Slightly thick, smooth	Moderate sweet, tempeh	Strong tempeh and palm sugar	
F2	H0	Dark brown	Slightly runny, smooth	Fairly sweet, tempeh	Moderate tempeh and palm sugar	Not acceptable
	H1	Light brown	Slightly runny, smooth, separated water	Fairly sweet, tempeh	Moderate tempeh and palm sugar	
	H2	Light brown	Slightly runny, smooth, separated water	Fairly sweet, tempeh	Moderate tempeh and palm sugar	
	H3	Light brown	Slightly runny, smooth, separated water	Fairly sweet, tempeh	Moderate tempeh and palm sugar	
F3	H0	Dark brown	Runny smooth	Fairly sweet, tempeh	Slightly tempeh and palm sugar	Not acceptable
	H1	Light brown	Runny smooth, separated water	Fairly sweet, tempeh	Slightly tempeh and palm sugar	
	H2	Light brown	Runny smooth, separated water	Fairly sweet, tempeh	Slightly tempeh and palm sugar	
	H3	Light brown	Runny smooth, separated water	Fairly sweet, tempeh	Slightly tempeh and palm sugar	
F4	H0	Yellowish brown	Slightly thick, smooth	Mildly sweet, tempeh	Strong tempeh and honey	Acceptable
	H1	Yellowish brown	Slightly thick, smooth	Mildly sweet, tempeh	Strong tempeh and honey	
	H2	Yellowish brown	Slightly thick, smooth	Mildly sweet, tempeh	Strong tempeh and honey	
	H3	Yellowish brown	Slightly thick, smooth	Mildly sweet, tempeh	Strong tempeh and honey	
F5	H0	Yellowish brown	Slightly runny, smooth, separated water	Fairly sweet, tempeh	Moderate tempeh and honey	Not acceptable
	H1	Yellowish brown	Slightly runny, smooth, separated water	Fairly sweet, tempeh	Moderate tempeh and honey	
	H2	Yellowish brown	Slightly runny, smooth, separated water	Fairly sweet, tempeh	Moderate tempeh and honey	
	H3	Yellowish brown	Slightly runny, smooth, separated water	Fairly sweet, tempeh	Moderate tempeh and honey	
F6	H0	Yellowish brown	Runny, smooth	Fairly sweet, tempeh	Slight tempeh and honey	Not acceptable
	H1	Yellowish brown	Runny, smooth, separated water	Fairly sweet, tempeh	Slight tempeh and honey	
	H2	Yellowish brown	Runny, smooth, separated water	Fairly sweet, tempeh	Slight tempeh and honey	
	H3	Yellowish brown	Runny, smooth, separated water	Fairly sweet, tempeh	Slight tempeh and honey	

**Table 6 foods-12-02789-t006:** Sensory evaluation results of mung bean tempeh ONS shots.

Formulation	Color	Taste	Aroma	Viscosity	Overall
F1	4.56 ± 0.21 ^b^	3.79 ± 0.24 ^b^	4.56 ± 0.22 ^a^	4.82 ± 0.20 ^a^	4.36 ± 0.21 ^b^
F4	5.77 ± 0.15 ^a^	3.55 ± 0.22 ^b^	4.54 ± 0.22 ^a^	4.67 ± 0.22 ^a^	4.23 ± 0.19 ^b^
CF	4.08 ± 0.21 ^b^	5.61 ± 0.20 ^a^	4.90 ± 0.21 ^a^	4.23 ± 0.26 ^a^	5.13 ± 0.18 ^a^

Different superscript letters indicate statistical significance (*p* > 0.05); Values are presented as mean ± SEM. A seven-point hedonic scale is used to rate five sensory attributes.

**Table 7 foods-12-02789-t007:** Composition of mung bean tempeh ONS one shot formulas.

Ingredients	F1		F4		Nutrition
	Amount	Calorie	Amount	Calorie	
Mung bean tempeh powder	5.20	19	5.20	19	2.04 g protein 2.45 g carbohydrate 0.17 g fat
Palm sugar	5.20	20	-	-	5.20 g carbohydrate
Honey	-	-	5.20	16	3.97 g carbohydrate
Olive oil	1.90	17	1.90	17	1.90 fat
Total	12.30	56	12.30	52	-

**Table 8 foods-12-02789-t008:** Calorie content of mung bean tempeh ONS one shot formulas.

Content	F1 (% kcal)	F4 (% kcal)	ESPEN Guidelines (% kcal)
Protein	14.30	15.60	8–30
Carbohydrate	53.60	49.20	30–65
Fat	32.10	35.20	25–40

## Data Availability

The data used to support the findings of this study can be made available by the corresponding author upon request.

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
