# Peer review of "Branched Chain Amino Acid Content and Antioxidant Activity of Mung Bean Tempeh Powder for Developing Oral Nutrition Supplements"

_foods, 2023, doi:10.3390/foods12142789_

Round 1

Reviewer 1 Report

Branched Chain Amino Acid Content and Antioxidant Activity of Mung Bean Tempeh Powder for Developing Oral Nutrition Supplement

The article proposes an innovative product oral nutrition supplement that is based on mung tempeh powder. I confirm the innovativeness of the research idea however, attempts to obtain a protein isolate from the mung tempeh bean were carried out in other study (Aspiyanto et al. 2019)  that should be mentioned here. In materials and methods some aspects need to be explained in more detail as given below.

Introduction

The introduction is well structured, but it is worth mentioning the changes in the content of anti-nutrients during the fermentation of mung bean tempeh and other beans (Abu-Salem et al. 2014).

Please also mention other studies in which attempts were made to obtain protein isolates from mung tempeh and what was the scope of these studies. It is worth highlighting what is new in your research.

Materials and methods

Material and methods need more explanation.

In section related to the Mung Bean Preparation and Tempeh Powder Production please indicate how much of mung beans already cooked without skins was mixed with 1g of the starter. Please indicate what type of bacteria was in the tempeh fermentation starter. Please tell me what the room temperature was exactly. Give more details about freeze-drying for example pressure or temperature during processing. Also specify the moisture content of the powder obtained after freeze-drying. What does food processor mean? Please specify the type of crusher?, model and manufacturer.

How was prepared non-fermented mung bean powder it has to be explain?

Most of the other methods lack types, models and manufacturers of devices used in the research, please complete this. Please specify exactly how many repetitions were made for all measurements

In the sensory test, please specify the age range of the panelists and how many men and women participated.

Results

In the result section, the author gives the results of the properties of mung bean tempeh powder and non-fermented mung bean powder, and the methodology does not specify how this non-fermented powder was made. There are big changes in the protein content in the results maybe this powder was not fermented from whole beans with skin without cooking it has to be explained because it's hard to understand.

Regarding Amino acids content in mung bean tempeh powder please add standard deviation values for every amino acid. How many replicates were done.

 Regarding Figure 4 all values are presented on one chart, so please adjust the letters denoting significant differences (a-d) not in one concentration, but referring to the whole so as to compare different concentrations

Proposed references:

Aspiyanto, Susilowati, A., Lotulung, P. D., Melanie, H., & Maryati, Y. (2019). Potency of stirred microfiltration cell in separation of fermented beans as protein isolate for natural source of folic acid. Indonesian Journal of Chemistry19(1), 9–18. https://doi.org/10.22146/ijc.25164

Abu-Salem, F. M., Mohamed, R. K., Gibriel, A. Y., & Rasmy, N. M. (2014). Levels of Some Antinutritional Factors in Tempeh Produced From Some Legumes and Jojobas Seeds. International Journal of Biological, Agricultural, Biosystems, Life Science and Engineering8(3), 280–285. Retrieved from

Reviewer 2 Report

The paper reports the formulation of a mung bean temph one shot  ONS with high in BCAA  content and high antioxidant activity. The work is generally carried out properly but some points should be addressed.

1. Table 4. For each fatty aid, please add the structural type (omega-3, omega-6, omega-9, saturated, unsaturated, etc). Please explain: linolenic  acid/omega-3 and linoleic acid/omega-6 (linolenic acid is omega-3; linoleic acid is omega-6). It is their ratio?

2.In the section 2.6 (Antioxidant activity assay), it is specified that the sampleswere diluted to a concentration of 0.5%. However, in the Results section (3.6; L277) it is mentioned that the highest antioxidant activity was noticed at a concentration of 5%. And also, the concentrations in the Figure 4 are 0.5; 1 and 5%. Please, clarify the values of used concentrations.

3. L372. We mentioned:  BCAA contains in ONS shot was 385.9 mg per 50 mL????

How does this value results?I think it is incorrect.

Mug bean  tempeh: total BCAAs = 54.14 mg/g

Mug bean tempeh powder content in ONS formulation: max. 10.4%

So: 281.528 mg BCAAs/50 mL formulation

Please, clarify this issue.

4. Conclusions can be reformulated after the corrections.

Reviewer 3 Report

The number of decimals must be uniformized in all manuscript.

Line 22: instead of continued.. reformulated the sentence, including ‘..were perfomed sensory analysis with..’

Line 33: minerals

Line 36-38: reformulated the sentence.. is not correct it..

Line 44: white and red beans are both common beans. So correct this information.

Line 81, 83..: insert ‘min’ to refer minutes.

Line 87, 89..: Insert ‘h’ to refer hours.

In the sub-section 2.2 proximate analysis, for protein and fat content determination is not necessary described exhaustively the methods, insert only the AOAC methods. You have to introduce the modifications made in relation of original method.

Line 130: for grammes is ‘g’ not ‘gr’.

Line 134: insert a space

Line 136: …mL… not …ml…

Line 194-195: mentioned the number of replications were performed for each analysis. Amino acids? Any data can not be published without statistical analysis, and for that at least triplicate are mandatory. So, you have to provide mean and SD of proximate composition, amino acids and fatty acids results.

Round 2

Reviewer 2 Report

The authors  answered to the questions and suggestion.
